# A three-dimensional finite element analysis on the effects of implant materials and designs on periprosthetic tibial bone resorption

Hyung Jun Park[1], Tae Soo Bae[2], Seung-Baik Kang[1]*, Hyeong Ho Baek[2], Moon Jong Chang[1], Chong Bum Chang[3]

1 Department of Orthopedic Surgery, Seoul National University College of Medicine, SMG-SNU Boramae Medical Center, Seoul, South Korea, 2 Department of Biomedical Engineering (BME), Jungwon University, Chungcheongbuk-do, South Korea, 3 Department of Orthopedic Surgery, Seoul National University College of Medicine, Seoul National University Bundang Hospital, Seongnam, South Korea

☯ These authors contributed equally to this work.

* ossbkang@gmail.com

## Abstract

### Introduction

Implant material is a more important factor for periprosthetic tibial bone resorption than implant design after total knee arthroplasty (TKA). The virtual perturbation study was planned to perform using single case of proximal tibia model. We determined whether the implant materials' stiffness affects the degree of periprosthetic tibial bone resorption, and whether the effect of material change with the same implant design differed according to the proximal tibial plateau areas.

### Materials and methods

This three-dimensional finite element analysis included two cobalt-chromium (CoCr) and two titanium (Ti) tibial implants with different designs. They were implanted into the proximal tibial model reconstructed using extracted images from computed tomography. The degree of bone resorption or formation was measured using the strain energy density after applying axial load. The same analysis was performed after exchanging the materials while maintaining the design of each implant.

### Results

The degree of periprosthetic tibial bone resorption was not determined by the type of implant materials alone. When the implant materials were changed from Ti to CoCr, the bone resorption in the medial compartment increased and vice versa. The effect of material composition's change on anterior and posterior areas varied accordingly.

**Data Availability Statement:** All relevant data are within the manuscript.

**Funding:** This work was supported by the Bio & Medical Technology Development Program of the

National Research Foundation (NRF) funded by the Korean government (MSIT) (2017M3A9D8063538).

**Competing interests:** The authors have declared that no competing interests exist.

## Conclusions

Although the degree of bone resorption was associated with implant materials, it differed depending on the design of each implant. The effect on the degree of bone resorption according to the materials after TKA should be evaluated while concomitantly considering design.

## Introduction

There have been reports that medial tibial periprosthetic bone marrow density (BMD) decreased after total knee arthroplasty (TKA) [1–3]. The medial tibial bone resorption after TKA may affect the outcome of TKA because the cancellous bone underneath the tibial base-plate mainly supports the component [3]. The bone resorption might compromise the stability of the component and may have the potential for aseptic loosening [2,4,5]. In terms of causes of periprosthetic bone resorption of the tibia, it is related to the stress shielding phenomenon [3, 6–9]. The factors associated with bone resorption caused by stress shielding could be classified into two factors: the patient factor such as preoperative varus deformity and body mass index (BMI), and the implant factor such as material composition, thickness of the baseplate, and design of the implant [7, 8, 10]. However, no consensus exits in clinical study, and thus, the implant factors associated with the resorption need to be further evaluated.

The stiffer component materials were reported to induce more stress shielding [7, 11, 12]. However, even with the same material, bone resorption was reported to be different depending on the different designs [7, 10]. One clinical study reported that the incidence and average amount of medial tibia bone resorption were greater with the thicker tibial baseplate than those with thinner one with same material at a minimum of 2 years after TKA (44% vs 10% and 1.07 mm vs 0.16 mm respectively) [7]. In contrast, another study revealed that there was no difference in the degree of bone resorption between two tibial components with the same materials but different designs [10]. Combining the previous findings, both implant design and material may be important to the degree of periprosthetic bone resorption. However, no study has reported that the extent to which each factor affects the degree of bone resorption. In order to investigate the inter-relationship between the material and design of the implant in terms of the degree of bone resorption, a research comparing implants made of different materials in the same design should be conducted. However, studies of these designs are difficult to conduct as clinical studies, so a research using finite element analysis may be an alternative.

The degree of periprosthetic bone resorption could differ according to the area of proximal tibia plateau. When the mechanical axis of lower extremity is from 1.1 to 1.5° varus alignment, the medial compartment of the proximal tibia is subjected to greater loading than the lateral compartment [13, 14]. Considering that the weight bearing line lies on the anteromedial proximal tibia and the reports of posteromedial side polyethylene wear after TKA, evaluation on both anteromedial and posteromedial areas may be more meaningful than other areas [15–17]. However, there is lack of information on the bone resorption pattern along the area of proximal tibial plateau.

We planned to perform virtual perturbation study using single case of proximal tibia model. The study aimed to determine (1) whether greater medial periprosthetic tibial bone resorption occur in CoCr than the titanium (Ti) tibial component, (2) whether the stiffness of implant materials affects the degree of periprosthetic tibial bone resorption under the same design, and (3) whether the effects of material change with the same implant design differ

according to the area of the proximal tibial plateau. We hypothesized that (1) the degree of medial periprosthetic tibial bone resorption would be greater in CoCr than Ti tibial component, (2) there would be more medial periprosthetic tibial bone resorption as the stiffness of implants increases under the same design, and (3) the degree of bone resorption would differ according to the areas of the proximal tibial plateau.

## Materials and methods

### Three-dimensional models of proximal tibia and tibial components

A three-dimensional (3D) finite element model of a proximal tibia was reconstructed through Mimics 20.0 (Materialize, Leuven, Belgium). To reflect the characteristics of in vivo proximal tibia consisted of cortical and cancellous bone, CT Hounsfield unit (HU) values was used in the reconstruction [9, 18–20]. The extracted computed tomography (CT) images as DICOM file from osteoarthritis patient (80 years old female, body mass index: 28.55kg/m$^2$, co-morbidity: hypertension, osteoporosis) were used to make the proximal tibia model. The proximal tibial plateau was divided into four areas based on the long and short axes of the interface of the proximal tibia and implant: anteromedial (AM), anterolateral (AL), posteromedial (PM), and posterolateral (PL) areas (Fig 1). To analyze the change in bone marrow density on proximal tibia, we included the area down to 5 mm from the resected surface.

Four commonly used posterior-stabilized (PS) type implants were included in the study: two (implant Ca, implant Cb) of them were made of CoCr and others (implant Ta, implant Tb) were made of Ti (Table 1). After being scanned by 3D scanner, the tibia components were implanted into the proximal tibia model perpendicular to the mechanical axis through 3-Matics 12.0 (Materialize, Leuven, Belgium). The introduced cement, with a thickness of 2 mm, between the tibia model and tibia implant was made through computer- aided design (CAD) with SolidWorks (Dassault Systems, Massachusetts, USA) [21]. Tibia components and cement were considered not to be displaced via fully bonded cement. The material properties of the component such as Young's modulus, Poisson's ratio values were set according to the data of the previous studies and have linear elastic, isotropic, and homogenous characteristics (Table 2) [9, 22, 23]. To determine the element size, element number, and node number, the convergence test was performed. The error in strain was found less than 3% when the element size of the component, cement and proximal tibia were less than 1mm. The mesh

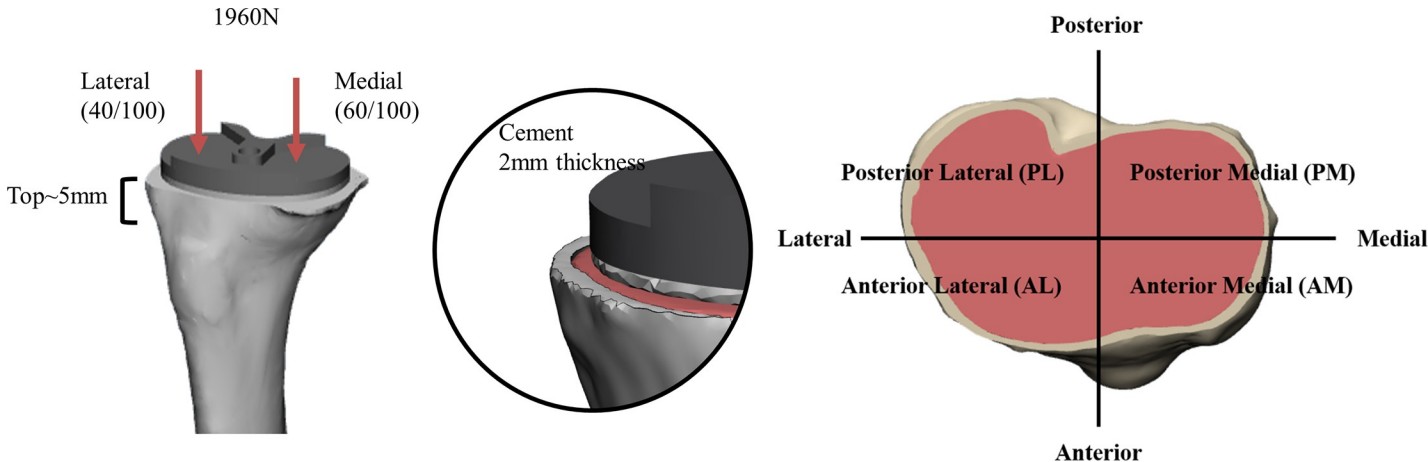

**Fig 1. Three-dimensional models of proximal tibia and tibial components.**

**Table 1. Material and design properties of tibial baseplates.**

| Properties | Implant Ca | Implant Cb | Implant Ta | Implant Tb |
|---|---|---|---|---|
| Material | Cobalt-Chrominum | Cobalt-Chrominum | Titanium | Titanium |
| Length of mediolateral | 72 | 71 | 72 | 70 |
| Length of anteroposterior | 48 | 46 | 50 | 50 |
| Thickness of baseplate (mm) | 4.2 | 4 | 2.6 | 3.4 |
| Slope of baseplate (°) | 0 | 0 | 3 | 5 |
| Design of baseplate | Symmetric | Symmetric | Asymmetric | Asymmetric |
| Length of stem (mm) | 42 | 46 | 50 | 38 |
| Design of stem | Tapered shape | Cruciate fin shape | Bar shape | Cylinder shape |

configuration what we used in mesh convergence test was described on Table 3. The post-processing was performed through Abaqus 14.0 (Dassault systems, USA)

The axial force of 1960 N was loaded on the point of implanted proximal tibia which was split 6:4 into medial and lateral compartment for normal gait [12, 21, 24]. The verbal informed consent was received from the patient for using the preoperative CT images to reconstruct the proximal tibial model. This was documented on the electrical medial record.

## Evaluation of strain energy density and the risk rate of bone resorption

The strain energy density (SED) was defined as the strain energy divided by the bone volume [25]. The area where the bone resorption and formation occur could be predicted by using SED [9, 25]. An increase of more than 75% in SED results in bone formation; otherwise, a decrease of more than 75% in SED which means implanted SED is less than 25% compared to original SED results in bone resorption [26].

$$Bone\ resorption: \left\{ \left( \frac{implanted\ SED}{original\ SED} \right) - 1 \right\} * 100 < -75\%$$

$$Bone\ formation: \left\{ \left( \frac{implanted\ SED}{origianl\ SED} \right) - 1 \right\} * 100 > 75\%$$

The proximal tibia was differentiated into the areas where at risk of bone resorption or formation with this formula. The areas with bone resorption and bone formation were indicated in red and blue colors, respectively (Fig 2). The proportion of the bone resorption hazardous area to the proximal tibia plateau could be calculated [9]. The proportion of bone resorption area was calculated with the original material composition of each implant and also assessed after exchanging the material composition from CoCr to Ti and vice versa.

**Table 2. Material properties of the components.**

| | Young's modulus (MPa) | Poisson's ratio |
|---|---|---|
| Implant | | |
| Implant Ca | 220,000 | 0.3 |
| Implant Cb | 220,000 | 0.3 |
| Implant Ta | 110,000 | 0.3 |
| Implant Tb | 110,000 | 0.3 |
| Cement | 2200 | 0.3 |
| Proximal tibia | $2017.3 \times \left( \frac{HU+13.4}{1017} \right)^{2.46}$ | 0.3 |

## Results

Although the implants Ca and Cb were made of the same CoCr materials, the implant Cb tended to show greater bone resorption than Ca. Moreover, the implant Ca even tended to show less bone resorption than the implants made of Ti (Table 3). There was the greatest bone resorption at medial compartment in the implant Cb (73.9%). However, the implant Ca showed the least bone resorption (60.7%). After the medial compartment being divided into anterior and posterior areas, the implant Cb showed the greatest bone resorption (60.2%) in the AM area and the second greatest bone resorption (87.0%) in the PM area after implant Tb. The implant Ca showed the lowest bone resorption (34.1%) in the AM area and the second lowest bone resorption (82.9%) after implant Ta in the PM area. Among all implants, the implant Ca had the least risk of bone resorption except only in the PM area (Fig 3) (Table 4).

When the materials of implants were changed from Ti to CoCr, the bone resorption in the medial compartment increases and vice versa. There was an increased bone resorption in implants Ta and Tb after exchanging the material from Ti to CoCr (0.1%, 1.3%, respectively). In contrast, the implants Ca and Cb showed decreased bone resorption (-1.3%, -0.8%, respectively) in the medial compartment (Fig 4) (Table 5).

The effect of the change in material composition on anterior and posterior areas varies according to the individual implants. After exchanging the materials, the implant Ca showed decreased bone resorption, whereas the implant Tb showed increased bone resorption in the medial compartment regardless of anterior and posterior areas. The change occurred mainly at the posterior in the implant Ca, while at the anterior in the implant Tb. The implants Cb and Ta showed a decreased and an increased bone resorption in PM area similar with the implants Ca and Tb. However, there were an increased and a decreased bone resorption at anterior area respectively (Fig 5). In contrast, there was an increased bone resorption in the lateral compartment regardless of materials after exchanging the material composition. The change of the implant Ca was the least and that of the implant Cb was the greatest (3.2%, 5.5%, respectively). In the overall area of the proximal tibia, the amount of bone resorption change in the implant Ca was the least and that of the implant Cb was the greatest (0.78%, 2.47% respectively). (Table 5)

## Discussion

Studies have reported medial periprosthetic tibial bone resorption after TKA using implants with stiffer materials [7, 9, 10]. Although medial periprosthetic tibial bone resorption is known

**Table 3. Mesh configuration.**

| Model | Part | Element number | Node number | Element type |
|---|---|---|---|---|
| Implant Ca | Tibial component | 570,744 | 94,886 | Tetraheadron |
| | Cement | 234,969 | 37,258 | Tetraheadron |
| | Proximal tibia | 673,939 | 93,420 | Tetraheadron |
| Implant Cb | Tibial component | 353,282 | 60,718 | Tetraheadron |
| | Cement | 109,995 | 23,063 | Tetraheadron |
| | Proximal tibia | 593,943 | 96,486 | Tetraheadron |
| Implant Ta | Tibial component | 384,527 | 65,825 | Tetraheadron |
| | Cement | 299,242 | 48,666 | Tetraheadron |
| | Proximal tibia | 677,607 | 93,584 | Tetraheadron |
| Implant Tb | Tibial component | 438,600 | 75,795 | Tetraheadron |
| | Cement | 259,499 | 46,692 | Tetraheadron |
| | Proximal tibia | 573,789 | 93,207 | Tetraheadron |

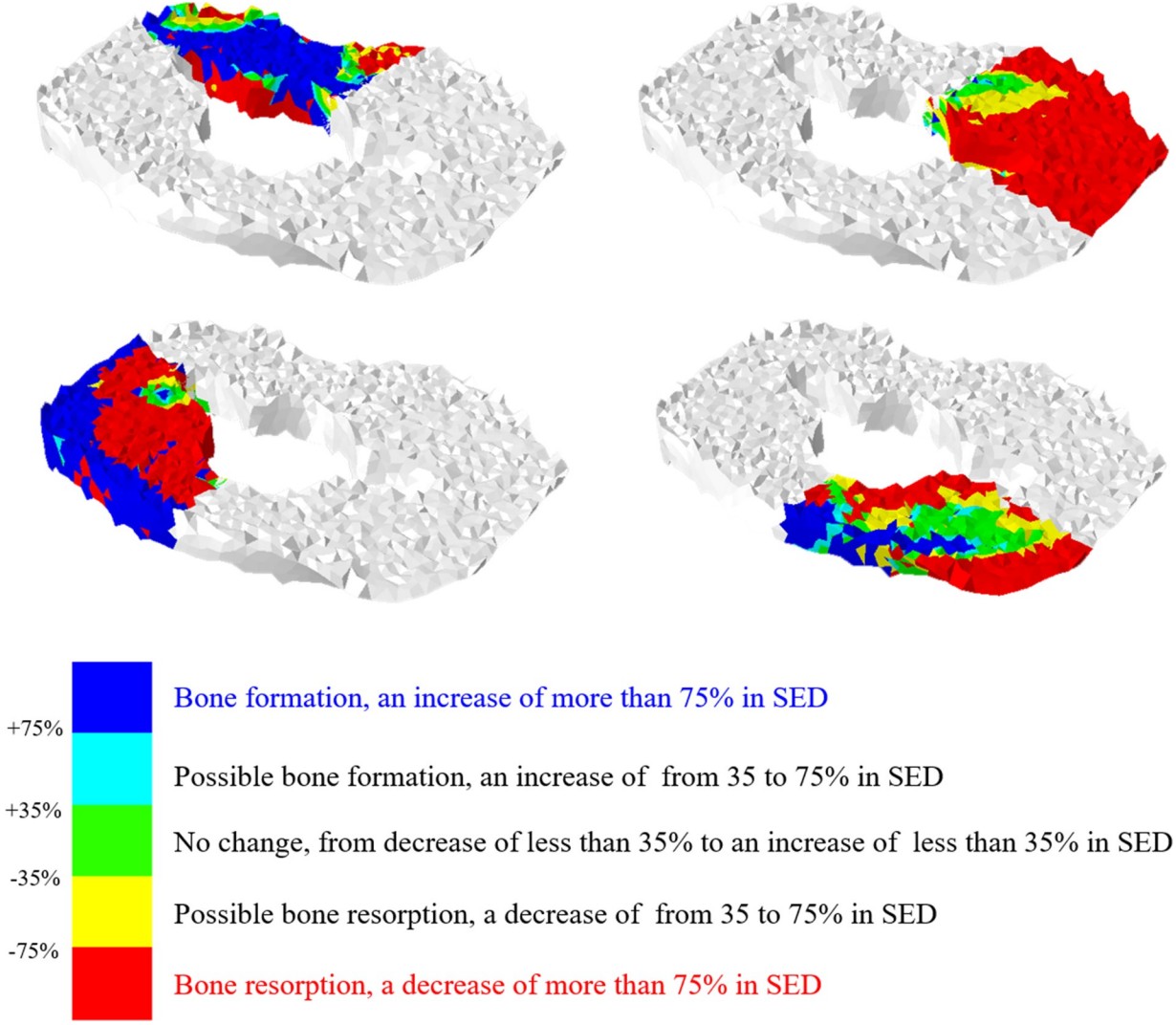

**Fig 2. Strain energy density (SED) and the bone resorption.**

to be related with stress shielding, there is no consensus on which factors are associated with stress shielding such as stiffness of the implant material, thickness of baseplate, and design of tibial component. The principal finding of our study was that although the stiffness of implant affected the degree of periprosthetic tibial bone resorption, it may be associated with other factors such as the design of tibial component.

The stiffness of implants did not always determine the bone resorption of each implant in our study which negated our hypothesis that the bone resorption was greater in CoCr than in the Ti tibial baseplate. Martin et al. compared the degree of bone resorption of three implants after TKA: CoCr, Ti and all polyethylene (AP) tibial baseplate. They reported that CoCr showed significant medial tibial bone loss (1.9 mm, 3.39 mm) compared to Ti (0.26 mm, 2.16 mm) and AP (0.05 mm, 1.24 mm) (average amount of defect in all patients, average amount of defect in only who had medial tibial bone loss, respectively). In this previous study, the medial tibial bone loss was evaluated by medial and lateral defect lengths on standing radiographs [7]. In another previous study, Yoon et al. conducted TKA with five implants: two CoCr and three

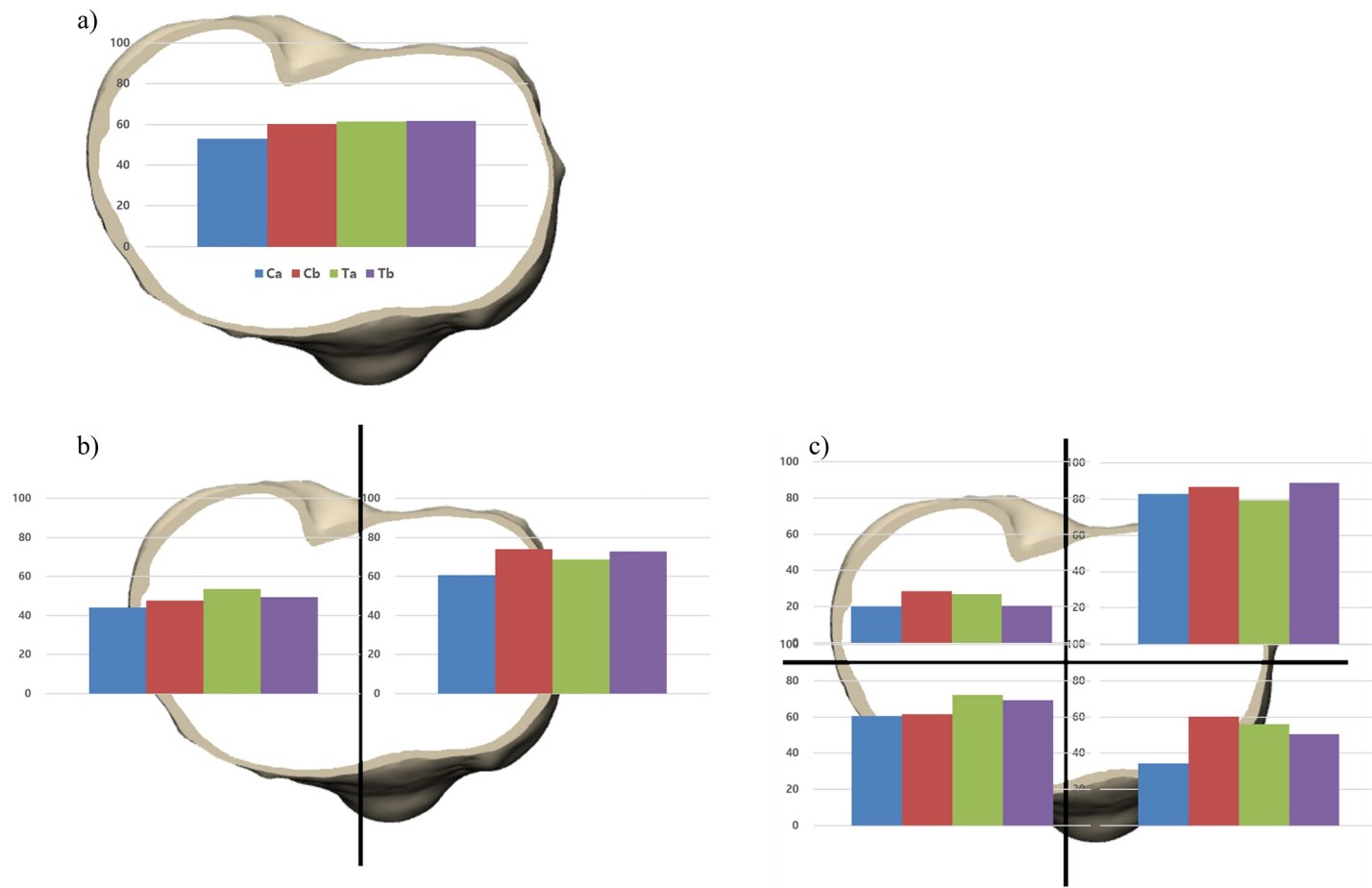

**Fig 3. The proportion of bone resorption area.** After implanting tibia baseplates, the proportion of bone resorption area was analyzed a) in overall area of proximal tibial plateau, b) in the medial and lateral compartment, and c) in the four areas of proximal tibia.

Ti tibial baseplates. They reported that CoCr showed more medial tibial bone resorption than Ti which was evaluated by the radiolucent line, bone mineral density (BMD) at medial tibial condyle. The incidence of medial tibial bone resorption was greater in CoCr than in Ti as 23.1% vs 7.9%, and the BMD decrease was also greater in CoCr than in Ti as 18.2% vs 13.1% (P < 0.05). Meanwhile, there was no difference in two CoCr tibia implants in terms of the bone resorption at two years after TKA [10]. Considering the latest concerns that CoCr tibial implants were associated with medial tibial bone marrow density [7, 10], our results appear

**Table 4. The proportion of bone resorption after implanting tibia baseplate.**

|  | Implant Ca | Implant Cb | Implant Ta | Implant Tb |
|---|---|---|---|---|
| Overall (%) | 53.0 | 60.3 | 61.4 | 61.6 |
| Medial (%) | 60.7 | 73.9 | 68.6 | 72.9 |
| Lateral (%) | 44.2 | 47.7 | 53.7 | 49.3 |
| Posteromedial (%) | 82.9 | 87.0 | 79.2 | 89.2 |
| Anteromedial (%) | 34.1 | 60.2 | 55.9 | 50.4 |
| Posterolateral (%) | 19.9 | 28.4 | 26.7 | 20.3 |
| Anterolateral (%) | 60.6 | 61.4 | 72.2 | 69.2 |

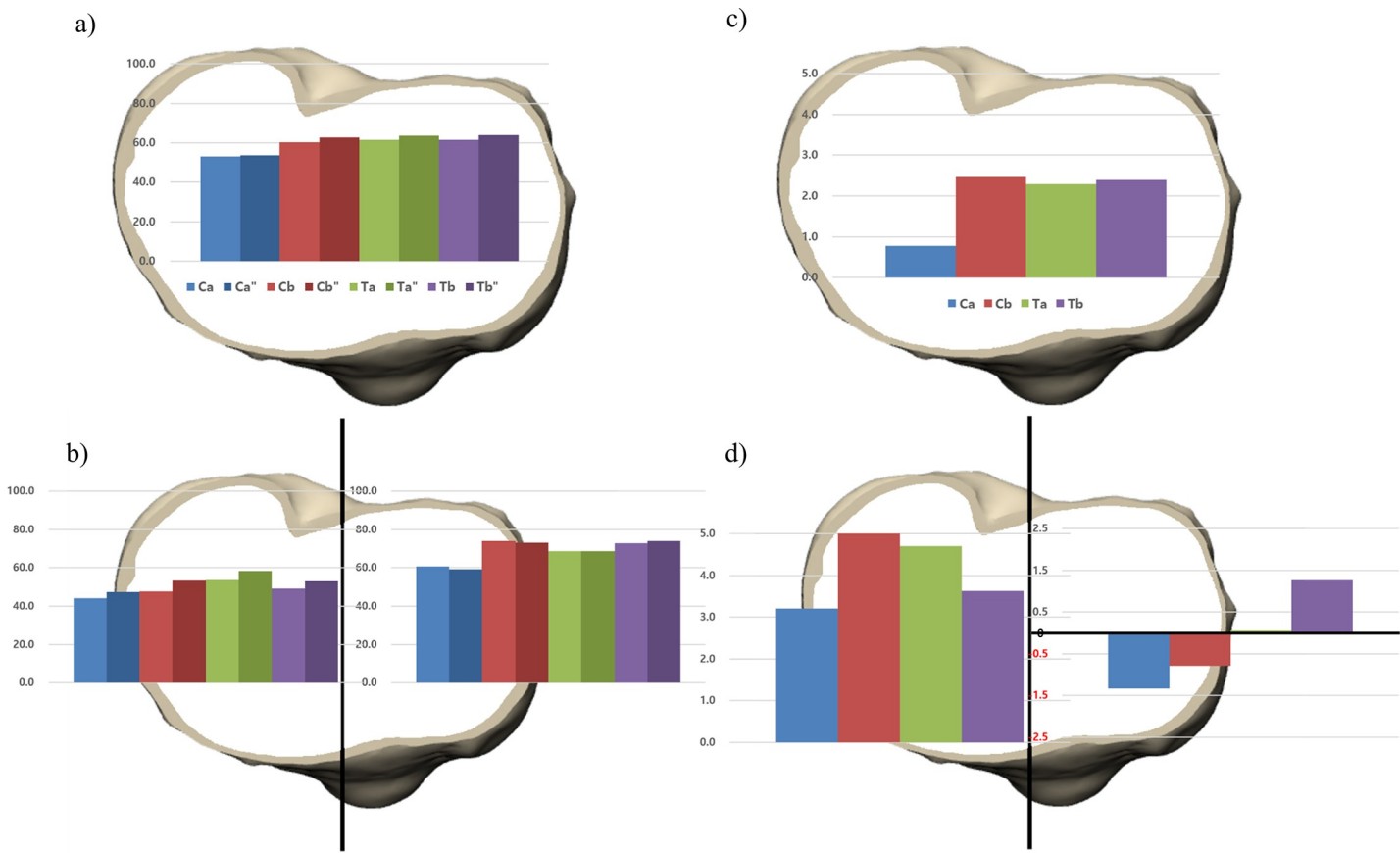

**Fig 4. The material composition of four implants were changed from cobalt-chromium (CoCr) to titanium (Ti) and vice versa while maintaining the design of the implant.** The proportion of bone resorption area after implanting four original and four altered tibial baseplates was analyzed in a) overall and b) medial and lateral compartment. The amount of bone resorption change was analyzed after changing the material composition in c) overall and b) medial and lateral compartment", the material composition of tibial baseplate is changed from CoCr to Ti.

different from those of aforementioned studies. This contradictory finding can be partly explained by two possible reasons. The parameters to assess the degree of bone resorption differed between the studies. The parameter, the proportion of bone resorption area, did not include the intensity of bone resorption but reflected whether each area was at risk of bone resorption or not, under the given conditions including the material composition, the design of implant, and thickness of tibial implant. The other possible reason is that the implant Ca used in our study is advanced version compared to the implant used in the previous studies

**Table 5. The amount of change in proportion of bone resorption after exchanging tibia implant from cobalt-chromium to titanium and vice versa.**

|  | Implant Ca | Implant Cb | Implant Ta | Implant Tb |
|---|---|---|---|---|
| Overall (%) | 0.8 | 2.5 | 2.3 | 2.4 |
| Medial (%) | -1.3 | -0.8 | 0.1 | 1.3 |
| Lateral (%) | 3.2 | 5.5 | 4.7 | 3.6 |
| Posteromedial (%) | -2.0 | -3.0 | 0.6 | 0.4 |
| Anteromedial (%) | -0.6 | 1.5 | -0.6 | 2.4 |
| Posterolateral (%) | 7.9 | 12.4 | 11.5 | 8.9 |
| Anterolateral (%) | 0.0 | 0.6 | 0.0 | 0.0 |

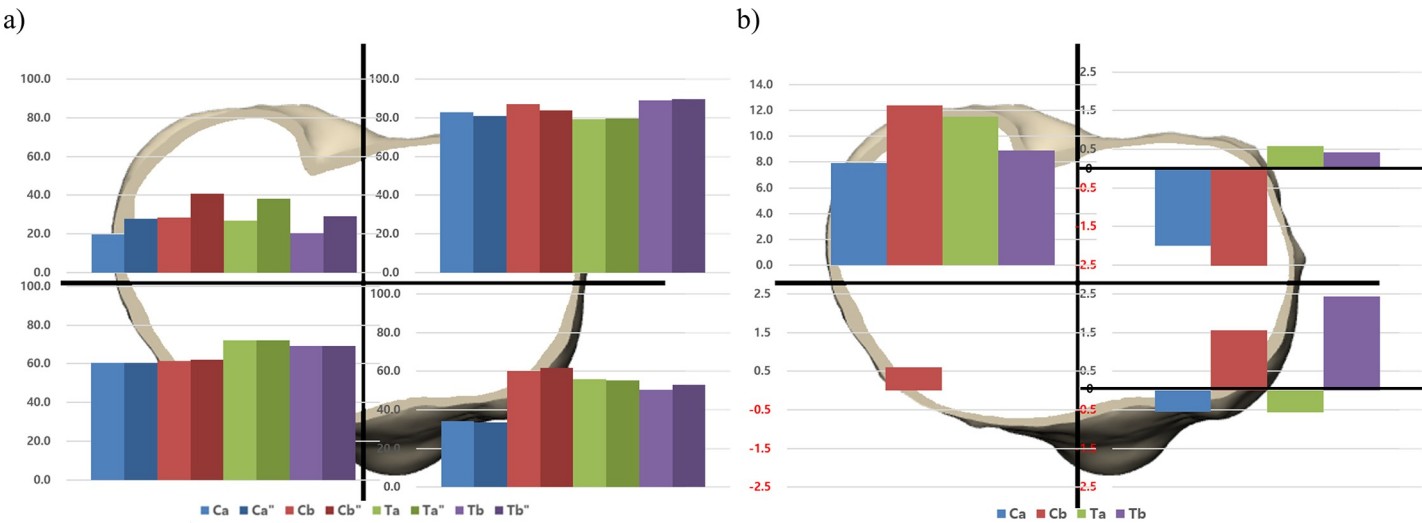

**Fig 5.** The proportion of bone resorption area after implanting four original and four altered tibial baseplates was analyzed in a) four areas in proximal tibia. The amount of bone resorption change after exchanging the material composition was analyzed in b) four areas in proximal tibia, the material composition of tibial baseplate is changed from CoCr to Ti.

[27, 28]. After the concerns of the implant Ca such as debonding were reported, the company made cement pocket under the tibial baseplate and increased roughness [27]. Unlike previous studies, the implant Ca showed significantly less periprosthetic tibial bone resorption compared to implant Cb despite consisting of the same CoCr materials. This may be the result of reflecting the design change. Therefore, not only the type of materials but also the design of component was should be considered in terms of the periprosthetic bone resorption.

Our results confirmed our hypothesis that there would be more medial tibial bone resorption as the stiffness of implants increases. There were investigations which reported that a stiffer implant induced more stress shielding [29]. Stress shielding of metal backed (MB) tibia baseplate was greater than that of the AP tibia baseplate [9, 30]. Among MB tibia baseplates, medial tibia bone resorption of CoCr was greater than that of Ti [7, 10]. Our findings were in line with that of previous studies that stiffer tibial implant caused more medial tibial bone resorption. In contrast, there was an increased bone resorption in the lateral compartment regardless of the tibial material composition. Hence, the tibial material composition affects medial and lateral compartments differently.

The effect of changing the material composition on anterior and posterior areas varies, proving our hypothesis, although the effect differs according to the implants. There was one study to report that high incidence of bone resorption on anterior portion of tibial baseplate than posterior portion and high incidence of bone resorption in tibial baseplate made of CoCr than that of Ti [10]. However, there was no study to report that the extent to which the bone resorption changed along the stiffness of the implant. In contrast to our expectation, there was an increased bone resorption in the implant Cb and a decreased bone resorption in the implant Ta at AM area after changing the material composition. The differences in bone resorption might be caused by the difference in the design of implants. In proximal tibia, the changing of material composition caused not only positive effects on some area but also negative effects on others especially in PM and PL areas. Interestingly, the changing of materials in the given designs had a negative effect on bone resorption. Based on these findings, we believe each design of the implant included in our study might be matched with proper material composition.

Several limitations of our study should be considered. First, there might be differences between experimental and clinical settings. In this study, we applied axial loading on proximal tibia only once. However, the result might be changed if we repeated applying the loading. Moreover, the pattern of load distribution on the surface of tibia would differ during range of motion of the knee joint. The experimental condition of this study was to use the loading condition to the standing state, and it does not reflect the change that occurs during range of motion. In addition, although the applied load on medial and lateral compartment was spilt into 6:4 considering a realistic condition from in vivo study, the load may be changed according to the activities. Nonetheless, we believe that this study can provide valuable information to readers as the material change experiment in the same implant design that is not possible in clinical studies

## Conclusion

Although the degree of bone resorption was associated with implant materials, the degree differed depending on the design of each implant. The changing of material composition in the same implant design affected anterior and posterior area on the plateau of proximal tibia differently according to the individual implant designs. The effect on the degree of bone resorption according to the materials after TKA should be evaluated while concomitantly considering design.

## Author Contributions

**Conceptualization:** Hyung Jun Park, Tae Soo Bae, Seung-Baik Kang, Hyeong Ho Baek, Moon Jong Chang, Chong Bum Chang.

**Data curation:** Hyung Jun Park, Tae Soo Bae, Hyeong Ho Baek, Moon Jong Chang.

**Formal analysis:** Hyung Jun Park, Tae Soo Bae, Seung-Baik Kang, Hyeong Ho Baek.

**Investigation:** Hyung Jun Park, Seung-Baik Kang.

**Methodology:** Hyung Jun Park, Tae Soo Bae, Hyeong Ho Baek, Moon Jong Chang, Chong Bum Chang.

**Supervision:** Tae Soo Bae, Seung-Baik Kang, Chong Bum Chang.

**Visualization:** Hyung Jun Park, Tae Soo Bae, Hyeong Ho Baek.

**Writing – original draft:** Hyung Jun Park, Tae Soo Bae, Hyeong Ho Baek, Moon Jong Chang, Chong Bum Chang.

**Writing – review & editing:** Hyung Jun Park, Tae Soo Bae, Moon Jong Chang.

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
