## [Decision Letter · Decision Letter 0]

23 Nov 2020

PONE-D-20-34669

A three-dimensional finite element analysis on the effects of implant materials and designs on periprosthetic tibial bone resorption

PLOS ONE

Dear Dr. Kang,

Thank you for submitting your manuscript to PLOS ONE. After careful consideration, we feel that it has merit but does not fully meet PLOS ONE’s publication criteria as it currently stands. Therefore, we invite you to submit a revised version of the manuscript that addresses the points raised during the review process.

We look forward to receiving your revised manuscript.

Kind regards,

Jose Manuel Garcia Aznar

Academic Editor

PLOS ONE

Journal Requirements:

2. Please provide additional details regarding patient consent for the CT images to be used for building the proximal tibia model in this research. In the ethics statement in the Methods and online submission information, please ensure that you have specified (1) whether consent was informed and (2) what type you obtained (for instance, written or verbal, and if verbal, how it was documented and witnessed).

Reviewers' comments:

Reviewer's Responses to Questions

**Comments to the Author**

1. Is the manuscript technically sound, and do the data support the conclusions?

Reviewer #1: Partly

2. Has the statistical analysis been performed appropriately and rigorously? 

Reviewer #1: N/A

3. Have the authors made all data underlying the findings in their manuscript fully available?

Reviewer #1: Yes

4. Is the manuscript presented in an intelligible fashion and written in standard English?

Reviewer #1: Yes

5. Review Comments to the Author

Reviewer #1: Many thanks for the consideration to review this interesting article proposal. It deals with a well as relevant topic of interest. Although my first suggestion would be to towards acceptance, please consider the concerns summarize as follows.

General Suggestions:

1. Clearly point out that this is a N=1 virtual perturbation study in the abstract and the introduction as well.

2. Consider the recommendations given in

Erdemir, A., Guess, T. M., Halloran, J., Tadepalli, S. C., & Morrison, T. M. (2012). Considerations for reporting finite element analysis studies in biomechanics. Journal of biomechanics, 45(4), 625-633.

Viceconti, M., Olsen, S., Nolte, L. P., & Burton, K. (2005). Extracting clinically relevant data from finite element simulations. Clinical Biomechanics, 20(5), 451-454.

3. Are the bone and implant geometries available (for further research comparisons)?

4. Please list the material properties in detail: Young's modulus, Poisson number, etc.

5. Please elaborate on the mesh configuration of bone and implants.

6. Could you quantify the structural stiffness of the different implants (cross-sectional area, area moments, etc.)?

7. Assumptions such as full bonding of interfaces should be justified in the text.

8. Check English spelling.

Major concerns:

1. Reporting of modeling details is insufficient, this includes:

a) mesh configuration (number elements, nodes, element types, mesh convergence)

b) boundary conditions (only loading is described without describing application in detail [point loads?], displacement constraints are missing)

c) geometries are only partially described, missing structural stiffness, some size dimensions (name implant type and size?)

d) Simulation software (solver, post-processing) is not clear

e) Validation was not performed, this should be justified

f) Material properties should be listed in detail and not only referenced

Detailed comments

l.98 "heterogenetic"? Did you mean heterogeneous?

l.112-3 is redundant with l.128-30, although a reference was added, are this in-vivo measured loads?

l.142-3 is redundant with l.149ff;

Figure legends do not add much information, please rather describe the figures in more detail. Add additional information directly on the figures, by this you can avoid confusion.

l.156 should be in the discussion, please just list the results first, before summarizing it

l.216 Please rephrase: “Studies have reported medial periprosthetic tibial bone resorption after TKA using implants with stiffer materials

l.217-220 Do not forget the exchanging loading in an in-vivo situation. Although you considered a realistic 60/40 ratio, this is not always true and change according to the activities.

6. PLOS authors have the option to publish the peer review history of their article (what does this mean?). If published, this will include your full peer review and any attached files.

Reviewer #1: **Yes: **Philippe Moewis

---

## [Author Response · Author response to Decision Letter 0]

4 Jan 2021

-> We have reviewed the sample of the manuscript and authors affiliation form.

2. Please provide additional details regarding patient consent for the CT images to be used for building the proximal tibia model in this research. In the ethics statement in the Methods and online submission information, please ensure that you have specified (1) whether consent was informed and (2) what type you obtained (for instance, written or verbal, and if verbal, how it was documented and witnessed).

-> We have added the informed consent from the patient on the method section. (Line 151-153)

General Suggestions:

1. Clearly point out that this is a N=1 virtual perturbation study in the abstract and the introduction as well.

-> We totally agreed with the reviewer’s comment. we have added what the reviewer commented. (Line 25-26, Line 86-87)

2. Consider the recommendations given in

Erdemir, A., Guess, T. M., Halloran, J., Tadepalli, S. C., & Morrison, T. M. (2012). Considerations for reporting finite element analysis studies in biomechanics. Journal of biomechanics, 45(4), 625-633.

Viceconti, M., Olsen, S., Nolte, L. P., & Burton, K. (2005). Extracting clinically relevant data from finite element simulations. Clinical Biomechanics, 20(5), 451-454.

- > We thank the reviewer for the recommendation.

3. Are the bone and implant geometries available (for further research comparisons)?

-> We totally agreed with the reviewer’s comment. As the reviewer’s comment, we have added geometries on table 1. (Line 141-142)

4. Please list the material properties in detail: Young's modulus, Poisson number, etc.

-> We totally agreed with the reviewer’s comment. As the reviewer’s comment, we have added material properties on table 2. (Line 143-144)

5. Please elaborate on the mesh configuration of bone and implants.

-> We totally agreed with the reviewer’s comment. As the reviewer’s comment, we have added mesh convergence test (Line 133-137) and mesh configuration on table 3. (Line 145-146)

6. Could you quantify the structural stiffness of the different implants (cross-sectional area, area moments, etc.)?

-> We totally agreed with the reviewer’s comment. As the reviewer’s comment, we have added geometries on table 1. (Line 141-142)

7. Assumptions such as full bonding of interfaces should be justified in the text.

-> Thank you for your commenting. As the reviewer’s comment, we clarified that the components and cement were full bonded and not to be displaced. (Line 129-130)

8. Check English spelling.

-> Thank you for your commenting. We have carefully reviewed the manuscript.

Major concerns:

1. Reporting of modeling details is insufficient, this includes:

a) mesh configuration (number elements, nodes, element types, mesh convergence)

-> We totally agreed with the reviewer’s comment. As the reviewer’s comment, we have added mesh convergence test (Line 133-137) and mesh configuration on table 3. (Line 145-146)

b) boundary conditions (only loading is described without describing application in detail [point loads?], displacement constraints are missing)

-> We totally agreed with the reviewer’s comment. As the reviewer’s comment, we clarified that the components and cement were full bonded and not to be displaced. (Line 129-130). The axial force was applied on the point of proximal tibial model. (Line 149-151). 

c) geometries are only partially described, missing structural stiffness, some size dimensions (name implant type and size?)

-> We totally agreed with the reviewer’s comment. As the reviewer’s comment, we have added geometries on table 1. (Line 141-142)

d) Simulation software (solver, post-processing) is not clear

-> We totally agreed with the reviewer’s comment. As the reviewer’s comment, we have added what we used in post-processing program. (Line 137)

e) Validation was not performed, this should be justified

-> We totally agreed with the reviewer’s comment. We also thought that validation was important issue in the virtual perturbation study. However, as the study was performed using clinical images of CT, validation was difficult to be done. Therefore, we selected the convergence test as the tool of justifying. Because the error in strain was less than 3% in our study, we considered that the findings of our study was reasonable. (Line 133-137)

f) Material properties should be listed in detail and not only referenced

-> We totally agreed with the reviewer’s comment. As the reviewer’s comment, we have added material properties on table 2. (Line 143-144)

Detailed comments

l.98 "heterogenetic"? Did you mean heterogeneous?

-> We totally agreed with the reviewer’s comment. As the reviewer’s comment, we clarified what we wanted to express in the word ‘hetrogenetic’. Because the proximal tibia was consisted of cortical and cancellous bone, the structural material of the proximal tibia was not homogenous. To reflect the characteristics of in-vivo, the proximal tibia model was reconstructed using CT images. (Line 102-105). 

l.112-3 is redundant with l.128-30, although a reference was added, are this in-vivo measured loads?

-> We totally agreed with the reviewer’s comment. As the reviewer’s comment, we have deleted the redundant sentences.

l.142-3 is redundant with l.149ff;

Figure legends do not add much information, please rather describe the figures in more detail. Add additional information directly on the figures, by this you can avoid confusion.

-> We totally agreed with the reviewer’s comment. As the reviewer’s comment, we have deleted the redundant sentences.

l.156 should be in the discussion, please just list the results first, before summarizing it

-> We totally agreed with the reviewer’s comment. As the reviewer’s comment, we have re-positioned that in the discussion section. (Line 272-273)

l.216 Please rephrase: “Studies have reported medial periprosthetic tibial bone resorption after TKA using implants with stiffer materials

-> We totally agreed with the reviewer’s comment. As the reviewer’s comment, we have changed. (Line 236-237)

l.217-220 Do not forget the exchanging loading in an in-vivo situation. Although you considered a realistic 60/40 ratio, this is not always true and change according to the activities.

-> We totally agreed with the reviewer’s comment. we have added what the reviewer’s comment to our one of the limitations. (Line 303-305)

---

## [Decision Letter · Decision Letter 1]

28 Jan 2021

A three-dimensional finite element analysis on the effects of implant materials and designs on periprosthetic tibial bone resorption

PONE-D-20-34669R1

Dear Dr. Kang,

We’re pleased to inform you that your manuscript has been judged scientifically suitable for publication and will be formally accepted for publication once it meets all outstanding technical requirements.

Kind regards,

Jose Manuel Garcia Aznar

Academic Editor

PLOS ONE

Additional Editor Comments (optional):

Reviewers' comments:

Reviewer's Responses to Questions

**Comments to the Author**

1. If the authors have adequately addressed your comments raised in a previous round of review and you feel that this manuscript is now acceptable for publication, you may indicate that here to bypass the “Comments to the Author” section, enter your conflict of interest statement in the “Confidential to Editor” section, and submit your "Accept" recommendation.

Reviewer #1: All comments have been addressed

2. Is the manuscript technically sound, and do the data support the conclusions?

Reviewer #1: Yes

3. Has the statistical analysis been performed appropriately and rigorously? 

Reviewer #1: N/A

4. Have the authors made all data underlying the findings in their manuscript fully available?

Reviewer #1: Yes

5. Is the manuscript presented in an intelligible fashion and written in standard English?

Reviewer #1: Yes

6. Review Comments to the Author

Reviewer #1: (No Response)

7. PLOS authors have the option to publish the peer review history of their article (what does this mean?). If published, this will include your full peer review and any attached files.

Reviewer #1: No

---

## [Editor Report · Acceptance letter]

1 Feb 2021

PONE-D-20-34669R1 

A three-dimensional finite element analysis on the effects of implant materials and designs on periprosthetic tibial bone resorption 

Dear Dr. Kang:

I'm pleased to inform you that your manuscript has been deemed suitable for publication in PLOS ONE. Congratulations! Your manuscript is now with our production department. 

Kind regards, 

on behalf of

Dr. Jose Manuel Garcia Aznar 

Academic Editor

PLOS ONE